# The Impact of a High-Carbohydrate/Low Fat vs. Low-Carbohydrate Diet on Performance and Body Composition in Physically Active Adults: A Cross-Over Controlled Trial

**DOI:** 10.3390/nu14030423

**Published:** 2022-01-18

**Authors:** Nadine B. Wachsmuth, Felix Aberer, Sandra Haupt, Janis R. Schierbauer, Rebecca T. Zimmer, Max L. Eckstein, Beate Zunner, Walter Schmidt, Tobias Niedrist, Harald Sourij, Othmar Moser

**Affiliations:** 1Division of Exercise Physiology and Metabolism, Department of Sport Science, University of Bayreuth, 95440 Bayreuth, Germany; nadine.wachsmuth@uni-bayreuth.de (N.B.W.); sandra.haupt@uni-bayreuth.de (S.H.); janis.schierbauer@uni-bayreuth.de (J.R.S.); rebecca.zimmer@uni-bayreuth.de (R.T.Z.); max.eckstein@uni-bayreuth.de (M.L.E.); beate.zunner@uni-bayreuth.de (B.Z.); Walter.Schmidt@uni-bayreuth.de (W.S.); othmar.moser@uni-bayreuth.de (O.M.); 2Interdisciplinary Metabolic Medicine, Division of Endocrinology and Diabetology, Department of Internal Medicine, Medical University of Graz, 8036 Graz, Austria; harald.sourij@medunigraz.at; 3Clinical Institute of Medical and Chemical Laboratory Diagnostics, Medical University of Graz, 8036 Graz, Austria; tobias.niedrist@medunigraz.at

**Keywords:** high-carb diet, low-carb diet, physical activity, body composition, metabolism

## Abstract

Background: Recently, high-carbohydrate or low-carbohydrate (HC/LC) diets have gained substantial popularity, speculated to improve physical performance in athletes; however, the effects of short-term changes of the aforementioned nutritional interventions remain largely unclear. Methods: The present study investigated the impact of a three-week period of HC/low-fat (HC) diet followed by a three-week wash-out-phase and subsequent LC diet on the parameters of physical capacity assessed via cardiopulmonary exercise testing, body composition via bioimpedance analysis and blood profiles, which were assessed after each of the respective diet periods. Twenty-four physically active adults (14 females, age 25.8 ± 3.7 years, body mass index 22.1 ± 2.2 kg/m^2^), of which six participants served as a control group, were enrolled in the study. Results: After three weeks of each diet, VO_2peak_ was comparable following both interventions (46.8 ± 6.7 (HC) vs. 47.2 ± 6.7 mL/kg/min (LC; *p* = 0.58)) while a significantly higher peak performance (251 ± 43 W (HC) vs. 240 ± 45 W (LC); (*p* = 0.0001), longer time to exhaustion (14.5 ± 2.4 min (HC) vs. 14.1 ± 2.4 min (LC); *p* = 0.002) and greater Watt/kg performance (4.1 ± 0.5 W/kg (HC) vs. 3.9 ± 0.5 W/kg (LC); *p* = 0.003) was demonstrated after the HC diet. In both trial arms, a significant reduction in body mass (65.2 ± 11.2 to 63.8 ± 11.8 kg (HC) vs. 64.8 ± 11.6 to 63.5 ± 11.3 kg (LC); both *p* < 0.0001) and fat mass (22.7% to 21.2%; (HC) vs. 22.3% to 20.6% (LC); both *p* < 0.0001) but not in lean body mass or skeletal muscle mass was shown when compared to baseline. Resting metabolic rate was not different within both groups (*p* > 0.05). Total cholesterol and LDL-cholesterol significantly decreased after the HC diet (97.9 ± 33.6 mg/dL at baseline to 78.2 ± 23.5 mg/dL; *p* = 0.02) while triglycerides significantly increased (76 ± 38 mg/dL at baseline to 104 ± 44 mg/dL; *p* = 0.005). Conclusion: A short-term HC and LC diet showed improvements in various performance parameters in favor of the HC diet. Some parameters of body composition significantly changed during both diets. The HC diet led to a significant reduction in total and LDL-cholesterol while triglycerides significantly increased.

## 1. Introduction

Undoubtedly, carbohydrate and/or caloric restriction remains the most important cornerstone in nutrition interventions to reduce body weight and avoid/treat related diseases summarized as metabolic syndrome [1]. Over the last years, high-carbohydrate vs. low-carbohydrate (CHO) diet has been a heated matter of debate for athletes in different sports and evidence remains mostly discrepant. The heterogeneity of recommendations based on study findings can be mainly attributed to the broad variety of different levels of performance, the individuality of athletes, age, sex and type- and volume-specific exercise investigated [2].

Low-carbohydrate (LC) diets, subdividable into very-low and low-CHO diets are defined by a proportion of less than 10% (20–50 g) and 26% (<130 g) of total caloric intake and a compensatory shift to proteins and fat as the energy source [3]. Such diets are suggested to be mainly effective due to the physiological induction of ketogenesis, however, the induction of ketosis strongly varies on an individual level [4]. The induction of nutritional ketosis leads to a glycogen depletion to avoid hypoglycemia; subsequently, the production of ketones by mobilizing fat from the adipose tissue is used as a fuel. Approximately 48 h after CHO restriction, glycogen stores are nearly fully depleted, and gluconeogenesis is responsible for the regulation of glucose homeostasis to provide glucose for the central nervous system and the red blood cells. To produce energy, fatty acids are released into the blood and shifted to the muscle and the liver to provide energy. Ketones then serve as important energy distributors for mitochondria-containing tissues such as the brain or muscle [5]. Fasting ketosis should not induce metabolic acidosis as long as there is no coexisting insulin deficiency or triggering medication present (e.g., SGLT-2 inhibitors) [6]. Within the last decade, ketone supplements, socially hyped as “superfuel” to power the mitochondrial engine, have flooded the market sparking a controversial discussion of being an ergogenic substance; nevertheless, the results remain inconclusive, not clearly detailing a favorable effect of ketone supplements [7,8]. Notwithstanding, ketogenic diets have proven real-world safety and efficacy in environments of insufficient supply of corn/vegetables [9]; as a short-term measure to optimize physical performance, LC diets remain critically scrutinized in the majority of evidence [10,11]. To date, no clear evidence is available indicating that an LC diet positively impacts high-intensity, strength, or power-based sports. Some studies suggested that LC might be able to delay exhaustion [12] during low-intensity prolonged exercise sessions and contribute to fat loss without compromising exercise performance [13,14,15]. The ability to spare muscle glycogen has been shown to potentially improve the ultra-endurance exercise performance [16]. However, the heterogeneity of study populations and settings is still a major factor for specific outcomes as, for example, individuals with obesity benefited from an LC diet during aerobic exercise by means of reducing body weight and promoting fat oxidation [15], while detrimentally losing resting muscle glycogen and endurance performance in a cycling study when an LC diet was tested against a balanced diet [17].

In contrast, high-carbohydrate diets are comparable to Western/European diets defined as a CHO proportion of total energy intake exceeding 45% [18]. Depending on the type of carbohydrates consumed, it remains the general recommendation to fuel/enhance performance as CHO remains the only macronutrient that can be metabolized immediately in order to provide substrates for high-intensity exercise sessions [19].

In athletes, nutrition plays a pivotal role since performance is closely related to specific diets that should be individualized and, in some cases varied over the annual training cycle. Therefore, especially short-term dietary interventions (~6 weeks) within a training microcycle may play an important role to improve performance [20]. A typical example to manage performance in athletes would be “carbohydrate cycling”, which is defined as the sequential periodization of low- and high-carbohydrate intake days that acutely improves performance without any impact on selected markers of metabolism during a training cycle [21].

However, generalized diets may also have health effects that vary in their efficacy based on individual responses. Hence, recommendations should be made with caution and need to be closely assessed whether specific guidance on macronutrients is suitable for this type of athlete.

From that point of view, it remains to be investigated how short-term changes in low vs. high carbohydrate diets influence performance in healthy and physically active people. Consequently, the aim of the present study was to investigate the impact of LC vs. HC diets on functional performance defined as peak oxygen uptake (VO_2peak_) and peak performance (P_peak_) as assessed during cardio-pulmonary exercise testing. As secondary outcomes, the influence of specific diets on body composition and metabolic blood parameters was investigated.

## 2. Materials and Methods

### 2.1. Design and Participants

This single-center, non-randomized, cross-over, controlled pilot trial was performed according to the declaration of Helsinki and the good clinical practice guidelines and was approved by the ethics committee of the University of Bayreuth, Germany (Ethics number: Az.O1305/1-GB/26042021). The study was conducted in conformity with the currently required local COVID-19 regulatory policy. Participants were between 18 and 41 years old, with normal nutritional behavior and had a body mass index of 18–27 kg/m^2^. The distribution of participants to the intervention or control group was substantially influenced by the COVID-19 pandemic in which official orders partly necessitated keeping the distance from the study center. For this reason, the first block of participants was supposed to participate in the control arm. Due to the fact that the study population consisted of sports students who participated in their studies during the semester, we had to agree on this procedure as timelines for study termination were limited.

### 2.2. Diet Intervention

For 3 weeks, participants who were allocated to the intervention group pursued a HC diet according to the macronutrient composition: 75–80% carbohydrates, 15% proteins and 5–10% fat. The high carbohydrate content was achieved with complex carbohydrates, such as found in whole meal products, potatoes or brown rice, while carbohydrates consumed via sucrose and fructose were mainly avoided. After a wash-out period of approximately 3 weeks, the second intervention period started with a one-week lead-in phase of a LC diet (20–25% carbohydrates, 15% proteins, 60–65% fat) followed by a 2-week ketogenic diet (5–7% carbohydrates, 15% proteins, 80% fat). The diet consisted mainly of fish, meat, nuts, vegetables and dairy products. Participants were educated by a nutritionist what to eat during the HC and LC diets and example menus were provided. Additionally, they were instructed to keep the total calorie intake stable during both interventions. The adherence to the specific diet was verified by diet diaries, which were documented throughout both study periods for at least 1 week during each intervention. The control group followed an identical schedule of investigations but was not supposed to adhere to any nutritional advice and was allowed to eat and drink ad libitum. The study procedure is illustrated in Figure 1.

### 2.3. Aerobic Performance

A cardio-pulmonary exercise (CPX) test was performed on a cycle ergometer (Excalibur, LODE^®^, Groningen, The Netherlands) to determine VO_2peak_, P_peak_ and Oxygen Efficiency after the period of 3 weeks of HC and LC eating, respectively. After a 3-min warm-up phase at 50 W, the load was continuously increased by 17, 17, or 16 Watts every minute (50 W/3 min) until volitional exhaustion. Respiration was analyzed breath-by-breath using the METALYZER^®^ 3B (Cortex, Leipzig, Germany) and VO_2peak_ was calculated as the mean value across the last 30 s before exhaustion. At rest, every 3 min during exercise, immediately at exhaustion and 1, 3, 5 and 7 min after exercise, capillary blood samples were taken from the earlobe to quantify lactic acid and blood glucose concentrations (Biosen S-Line, EKF-Diagnostic, Barleben, Germany) [22].

### 2.4. Resting Metabolic Rate

Resting metabolic rate (RMR), also performed after the two diet-periods, was measured in the morning after a 12-h overnight fast by indirect calorimetry using breath-by-breath technology (METALYZER^®^ 3B, Cortex, Leipzig, Germany) after each of the diet-periods. Measurements took place with the participant resting in a supine position for 30 min and were performed in a laboratory. The temperature and environmental humidity in the laboratory of the research facility were stable during all visit days with 24 °C and 50%, respectively. A steady state of 5 min was used to calculate RMR. 

### 2.5. Body Composition

At the screening visit and weekly during each trial arm, measurements of body mass, fat mass, skeletal muscle mass and total body water were performed using bioelectrical impedance analysis (InBody 720, JP Global Markets GmbH, Eschborn, Germany). Measurements were performed in a fasted and undressed state following the standardized specifications of the manufacturer.

### 2.6. Daily Activity and Nutrition

Daily physical activity was assessed via a continuous Bluetooth^®^ activity monitor (ActiGraph wGT3X-BT, ActiGraph^®^, Pensacola, FL, USA) and an activity diary for a total of one week during both diet interventions, respectively. The ActiGraph was placed on the hip throughout the day with recording breaks, e.g., showering, being documented in the diary. The activity data collected by the activity tracker were analyzed using the ActiLife 6 software (ActiGraph^®^, Pensacola, FL, USA). The Freedson VM3 (2011) formula was used to calculate the activity calories of the subjects.

Simultaneously, a food diary was used to monitor the composition of macronutrients in both interventions in a one-week interval. Diaries were analyzed using PRODI^®^ 6.5 Expert-Version (Nutri-Science GmbH, Hausach, Germany).

### 2.7. Venous Blood Samples

Venous blood samples were obtained from the antecubital vein at baseline and after three weeks of the respective diet interventions.

The blood serum vacutainer was left to rest for a minimum of 30 min prior to being centrifuged at room temperature for 10 min at 1500× *g*. The serum was then aliquoted and stored at −80 °C at the research facility. Once the study was completed serum samples were analyzed for betahydroxybutyrate, lipids (total cholesterol, HDL/LDL, triglycerides), c-reactive protein (CRP) and interleukin-6 (IL-6). All the listed measurements were performed on a cobas 8000 analyzer (Roche Diagnostics, Basel, Switzerland) with standardized assays by the same manufacturer, calibrated to international standards.

### 2.8. Statistical Analysis

All data were assessed for normal distribution by means of the Shapiro-Wilk normality test. Performance parameters, hematological, physical activity and nutrition parameters were analyzed via repeated measures two-way analysis of variance with Geisser-Greenhouse correction. Differences between groups, timepoints and group x timepoint were calculated in the same manner. Sidak’s multiple comparisons test with individual variances was computed for each comparison. Differences between groups were calculated via paired t-test. Correlations were conducted via Pearson tests between VO_2peak_, P_peak_ and time to exhaustion (TTE) as the dependent variables while nutrition and blood parameters were the independent variables.

All data were calculated via GraphPad Prism Version 8.0.2 (GraphPad Software, Inc., San Diego, CA, USA). Statistical significance was accepted at *p* < 0.05.

## 3. Results

In total, 24 individuals participated in the study, 18 of them (13 females, mean age 24.9 ± 1.3 years, BMI 21.8 ± 1.8 kg/m^2^) took part in the intervention group (high-carb, low-carb and ketogenic diet) while 6 of the participants (1 female, mean age 28.5 ± 6.9 years, BMI 23.1 ± 1.7 kg/m^2^) served as control group, who adhered to usual nutritional behavior. As mentioned above, an appropriate randomization was not possible due to the restrictions, which COVID-19 made abruptly necessary. This also resulted in a heterogeneous distribution of study participants to the respective study arms. All screened participants were eligible to participate in the study in which no participant had to be withdrawn or left the study prematurely. Further baseline characteristics are given in Table 1.

### 3.1. Performance Parameters 

#### 3.1.1. Peak Oxygen Uptake (VO_2peak_)

After three weeks of HC and LC diets, a CPX test until volitional exhaustion was conducted. No significant differences were found between both diets with respect to the relative VO_2peak_: 46.8 ± 6.7 (HC) vs. 47.2 ± 6.7 mL/kg/min (LC; *p* = 0.58). Furthermore, no significant difference was seen when comparing the control group (49.4 ± 7.4 (timepoint 1) and 47.2 ± 9.4 (timepoint 2) to HC (timepoint 1, *p* = 0.87; timepoint 2, *p* = 0.99) and LC (timepoint 1, *p* = 0.91; timepoint 2, *p* = 0.99)).

#### 3.1.2. Peak Performance (P_peak_)

During the CPX test, we found a significantly higher P_peak_ with 251 ± 43 W in the HC arm when compared against 240 ± 45 W in the LC arm (*p* = 0.0001). Furthermore, P_peak_/kg was significantly higher in HC diet with 4.1 ± 0.5 W/kg versus 3.9 ± 0.5 W/kg in the LC arm (*p* = 0.003). In the control group the P_peak_ at both time points was significantly higher when compared to HC (*p* = 0.02 (time point 1) and 0.03 (time point 2)) and LC (*p* = 0.009 (time point 1) and 0.02 (time point 2)). 

#### 3.1.3. Oxygen Efficiency (O_2_ Efficiency)

Oxygen consumption was not different at volitional exhaustion (Figure 2A). However, due to higher P_peak_ in the HC arm oxygen consumption per Watt was significantly higher after the LC diet when compared to HC diet at 250 W (12.4 ± 0.9 mL/W versus 11.7 ± 1.0 mL/W; *p* = 0.02) and at exhaustion (12.1 ± 1.2 mL/W versus 11.4 ± 1.3 mL/W; *p* = 0.003) (Figure 2B).

#### 3.1.4. Time to Exhaustion (TTE)

TTE was significantly longer after the HC diet when compared to LC diet (14.5 ± 2.4 min versus 14.1 ± 2.4 min; *p* = 0.002).

TTE was also significantly longer when comparing both time points of the control group vs the LC and HC diet: (18.2 ± 3.4 min at time point 1 (*p* = 0.01 for LC and *p* = 0.02 for HC) and 18.1 ± 4.1 at time point 2 (*p* = 0.009 for LC and *p* = 0.04 against HC)). Data on physical activity parameters according to the group and group comparisons are shown in Table 2. 

#### 3.1.5. Heart Rate during CPX Test

No significant differences were found between HC and LC, as well as in the control group for heart rate during CPX testing (Figure 3A).

#### 3.1.6. Blood Lactate during CPX Testing

No significant differences were found between HC and LC for blood lactate levels during CPX testing. When compared to the control group, blood lactate was significantly increased after HC diet when compared to both CPX tests during the control phase 1 (Figure 2) (12 min: *p* = 0.03, 15 min: *p*= 0.002, 18 min: *p* < 0.0001) and control phase 2 (12 min *p* < 0.0001, 15 min: *p* < 0.0001, 18 min: *p* < 0.0001). No difference in LC and control phase 1 was found but in control phase 2 (12 min: *p* = 0.02, 15 min: *p* = 0.03, 18 min: *p* = 0.02) compared to LC (Figure 3B).

#### 3.1.7. Blood Glucose during CPX Testing

HC showed significantly higher blood glucose levels at timepoint 0 (*p* = 0.005) and timepoint 3 (*p* = 0.03) compared to LC. No significant differences between groups or control phases were found throughout the exercise tests (Figure 3C).

#### 3.1.8. Correlations of Respiratory and Functional Capacity Data

Correlations conducted for VO_2peak_, P_peak_ and TTE found only total calorie intake as a significant parameter. It correlates with VO_2peak_ (r^2^ = 0.35, *p* = 0.002), P_peak_ (r^2^ = 0.35, *p* = 0.0009) and TTE (r^2^= 0.35, *p* = 0.0008).

### 3.2. Body Composition, Physical Activity and Nutrition

#### 3.2.1. Body Mass

HC diet led to a significant decrease in body mass from 65.2 ± 11.2 kg at baseline to 63.8 ± 11.1 kg after 3 weeks (*p* < 0.0001). In the LC arm, body mass also decreased significantly compared to baseline: 64.8 ± 11.6 to 63.5 ± 11.3 kg (*p* < 0.0001) (Table 3). No significant changes were found when LC was compared against HC (*p* = 0.99) and when the control group was compared to both intervention arms (*p* > 0.05).

#### 3.2.2. Lean Body Mass and Skeletal Muscle Mass

Following the dietary interventions and measurements at baseline (timepoint 0) in comparison to timepoints after the 3-weeks of the respective diet, no significant changes in lean body mass and skeletal muscle mass were found for HC and LC (*p* > 0.05) (Table 3). No significant changes were also found when the control group was compared with both intervention arms (*p* > 0.05).

#### 3.2.3. Body Fat and Visceral Fat

A significant decrease in body fat percentage was found for HC from baseline (22.7%) to week 3 after the specific diet (21.2%, *p* < 0.0001). In the LC group also, a significant decrease was found from baseline (22.3%) to the end of the LC diet after 3 weeks (20.6% *p* < 0.0001). Visceral fat decreased in HC from baseline (58.56 cm^2^) to (53.91 cm^2^ after the HC diet (*p* < 0.0001). In addition, visceral fat decreased from baseline (57.58 cm^2^) until the end of LC (50.55 cm^2^; *p* < 0.0001) (Table 3). No significant changes were found when the control group was compared with both intervention arms (*p* > 0.05). 

#### 3.2.4. Physical Activity

During HC, the participants daily walked 7539 ± 2110 steps, during LC 6702 ± 2272 steps were performed, the control group walked 7771 ± 2703 steps during period 1 and 7262 ± 2599 during period 2 (*p* = 0.66). Ingested activity calories per day were also not significantly different (HC (423 ± 163 kcal), LC (363 ± 175 kcal), control group 1 (489 ± 172 kcal) and control group 2 (430 ± 186 kcal) (*p* = 0.43).

#### 3.2.5. Resting Metabolic Rate (RMR)

No significant difference was found for RMR (kcal/day) when comparing HC (1606 ± 267) vs. LC (1670 ± 302). Moreover, when compared to control phase 1 (1960 ± 449) and control phase 2 (1885 ± 461), no significant difference was seen (all *p* > 0.05).

#### 3.2.6. Nutrition

Caloric intake was significantly lower in HC compared to LC with 1739 ± 606 vs. 1939 ± 430 kcal/day (*p* = 0.02). Furthermore, average CHO intake was significantly higher in HC compared to LC with 74 ± 4 vs. 7 ± 2% of the entire daily energy intake. Consumed protein in % was significantly lower in HC compared to LC with 13.6 ± 2.3 vs. 22.1 ± 3.4% of daily intake. Fat intake was significantly lower in HC compared to LC with 9.8 ± 2.6 vs. 68.5 ± 5.5% of total energy intake. Further details on the distribution of macronutrients are given in Figure 4.

### 3.3. Biochemical Parameters

#### 3.3.1. Beta-Hydroxybutyrate

Ketone levels did not change during HC: 0.06 ± 0.05 vs. 0.05 ± 0.07 mmol/L (*p* = 0.99) while levels increased significantly in the LC group from 0.04 ± 0.02 mmol/L before to 0.42 ± 0.27 mmol/L in response to this diet (*p* < 0.0001). Ketone levels after the LC diet were significantly higher in comparison to ketones after the HC diet (*p* < 0.0001). Ketones in the control group did not change during the time of the study.

#### 3.3.2. Total Cholesterol 

Following the HC diet, a significant reduction of total cholesterol levels from baseline to the end was found (189 ± 34 versus 158 ± 27 mg/dL; *p* = 0.02); no such changes were seen during the LC diet (187 ± 35 vs ± 203 ± 60 mg/dL) or in the control group when comparing from baseline to follow-up (*p* > 0.05).

#### 3.3.3. Low-Density-Lipoprotein (LDL-C) 

From baseline to the end of the HC diet phase, LDL decreased from 97.9 ± 33.6 mg/dL to 78.2 ± 23.5 mg/dL (*p* = 0.02). No significant difference was seen after LC or when compared to the control group (all *p* = 0.21).

#### 3.3.4. High-Density-Lipoprotein (HDL-C)

HDL-C levels significantly decreased during HC with 77 ± 9 mg/dL at baseline to 58 ± 9 mg/dL after the diet intervention (*p* < 0.0001) while a not significant increase (77 ± 11 to 82 ± 916 mg/dL) of HDL was observed during LC diet (*p* = 0.34). Following an HC diet, HDL was lower than prior to LC (*p* = 0.0003) and after LC (*p* = 0.0002). No significant results were seen when compared to the control group.

#### 3.3.5. Triglycerides (TG) 

TG significantly increased during the HC diet (76 ± 38 to 104 ± 44 mg/dL; *p* = 0.005). No significant change of TG during LC diet or control was seen (*p* > 0.05).

#### 3.3.6. C-Reactive Protein (CRP) and Interleukin 6 (IL-6) 

Both dietary interventions had no influence on CRP or IL-6 -levels and intergroup comparisons showed no difference (*p* = 0.17). Changes of laboratory parameters according to the specific intervention arm are shown in Table 4.

## 4. Discussion

The present study investigated the effects of a high-carbohydrate versus low-carbohydrate diet in healthy individuals on physical performance, body composition and laboratory profiles. The primary objective was to determine differences in functional capacity parameters assessed by cardio-pulmonary exercise testing.

In this context, the achieved VO_2peak_ was comparable after both interventions while the time to exhaustion was superior when the HC diet was conducted. Of note, we did not observe major significant results when we compared performance data with the control group (except a significantly longer time to exhaustion in the control group when compared to the LC group). This finding and all the comparisons to the control group must be taken with caution as the control group was of a small sample size and characterized by heterogeneous baseline characteristics (higher age, higher weight). When considering the longer time to exhaustion during the HC diet compared to the LC diet, similar results have previously been shown by Burke et al., highlighting that already after three weeks of training on an LC diet, endurance performance deteriorates by 1.6% while on a high-carbohydrate diet it may improve by >6% [23]. The comparable TTE with the control group arises the question of whether HC itself or simply normal carbohydrate ingestion led to a better TTE when compared to an LC diet. In this matter, Pitsiladis and Maughan demonstrated that an HC diet (70% CHO) did not show a longer TTE when compared to a diet with a normal CHO proportion (40%) [24]. Furthermore, after the LC diet, we found a higher O_2_ efficiency with increasing performance with a peak at volitional exhaustion (+5.6%). That might impair high-intensity exercise due to reduced exercise economy, which is also reported in the review of Burke [25]. However, especially for professional athletes aiming to find the last mosaic pieces to improve performance, an acute HC diet might enhance endurance performance, especially stamina. Notwithstanding, such interventions must resonate with the regular training structure to find an optimal balance between exercise volume and macronutrient intake.

Based on this, it is also of particular interest to investigate the blood lactate kinetics measured during the CPX testing. Blood lactate values were not significantly different in our study in comparison to both diets at baseline—yet after 10 min of testing the blood lactate response of the LC group showed an altered kinetic, which is in line with previous findings from Hu et al. [26]. In their study, it was shown that LC diets reduce resting plasma lactate levels; however, if this finding impacts functional capacity and physiological markers need to be investigated during a longer period of carbohydrate-restricted feeding, larger cohorts and during varying types of exercise accompanied by different intensities.

Apart from a significant physiological rise in betahydroxybutyrate in the LC diet group, which implicates sufficient adherence to the diet measures, some alterations in the lipid profiles were observed. In our participants, who were all classified as normocholesterolemic, total cholesterol and LDL-C decreased significantly during the HC intervention, which is in line with previous research that found reductions in total and LDL cholesterol when normocholesterolemic participants switched to a diet low in saturated fat but high in carbohydrates [27]. In contrast, total cholesterol and LDL-C numerically increased during the LC diet without reaching statistical significance. This finding confirms results from a recent study, in which an LC/high-fat diet increased LDL- cholesterol by 44% with high interindividual variability of increase (5–107%) during a 4-week diet intervention [28]. The only slight increase of cholesterol in our study might be due to the fact that the study was conducted in well-trained individuals who regularly perform physical activity/exercise. The effect of this physically active lifestyle might attenuate the cholesterol increase induced by the LC/high-fat diet [29]. However, as seen for example in the DIRECT study, the effect of the LC diet on LDL-C is changing over time, hence the duration of the study seems to be an important factor to consider when evaluating the lipid effects of LC diets [30]. In our study, triglyceride levels significantly increased after the HC diet and tended to decrease without significance during the LC diet. The effect of different diets on triglyceride levels remain heterogeneous, although a reduction in triglycerides with LC is suggested in a meta-analysis in people with diabetes mellitus type 2 [3] and the phenomenon of carbohydrate-induced hypertriglyceridemia (HPTG) has been previously observed in studies where HC and low-fat diets were investigated. Data suggested that hepatic insulin resistance may be associated with an increased lipogenesis [31]; whether this increased lipogenesis is influenced by different types of sugars, whether diet success may be evaluated according to triglyceride levels and whether HPTG serves as a risk factor or prognostic biomarker for the metabolic disease remain a matter of research in future.

Data from the nutrition diaries proved a sufficient adherence to the predetermined nutritional measures as shown by a 74% and 7% distribution of CHO intake during HC and LC diets. Moreover, the changes in betahydroxybutyrate indicated efficacy in terms of achieving the ketogenic state during the LC phase. Additionally, physical activity during the two diet phases was not different, so that changes in the measured parameters are not considered to impact performance parameters.

Within our study, we detected some changes in anthropometric data in response to the specific diets. In detail, total body mass decreased significantly in both intervention arms within a short period of time (three weeks of HC and LC diet). Recently, meta-analyses have shown that weight reduction can be achieved with both of our performed diets [32] with a more pronounced weight loss when LC diet was conducted [33]. However, these studies were mostly performed for a longer period (at least six months) and thus achieved a higher impact on weight reduction, which contrasts with our short-term intervention. The observed weight reduction can be mainly declared by a significant decrease in proportional body fat and visceral fat. Previous studies have shown that visceral fat is closely related to metabolic consequences of obesity and excessive visceral fat is thought to release fatty acids in the portal vein inducing an increased hepatic fat accumulation. A rapid, yet even a small reduction in visceral body fat might have acute health benefits, yet does not directly influence functional performance, as seen in our study. Besides fat loss, the positive short-term effect of LC on body mass might be also explained by an intensified sodium diuresis, which is known to occur mainly in the early phases of low and in particular very low-carb diets [34]; however, in our study total body water did not substantially change during LC diet. Existing evidence on changes in anthropometric data in response to HC and LC diets remains highly heterogeneous and an active matter of debate. The partly confusing and discrepant results in regard to diet-induced weight effects gained from clinical trials might be justified by different durations of dieting, variable definitions of high and low carb diets as well as different populations which were investigated. In our point of view, each broadly accepted diet dedicated to reducing bodyweight has the potential to positively modify weight control when conducted as recommended, in particular when food diaries are kept and nutritional support by experts is guaranteed, as was the case in our study. Considering this, a crucial feature of successful dieting includes positive encouragement and individual interest in strictly adhering to the suggested nutritional behavior. Intriguingly, when evaluating bodyweight reduction, the diet-unspecific “adherence- and support-effects” might overcome the imbalanced kcal/day intake in comparison of both diet interventions; even though within the HC arm fewer kcal/day were consumed, the bodyweight reduction was nearly congruent for HC and LC. From our point of view, a generalized diet prescription to reduce the bodyweight by means of HC or LC might be outdated and the focus should be shifted preferably towards a personalized dietary prescription.

Typical markers of inflammation (CRP, IL-6) were not altered in response to the two different diets. Considerably, IL-6 which serves as a key cytokine triggered by systemic inflammation has shown also to have a major correlation to acute and strenuous physical activity and might have lipolytic properties during a ketogenic state. Our study indicates no such effects when assessed without recent exercise and after reaching significant ketosis after extreme carbohydrate restriction [35].

Of note, we must address moderate limitations in our study: firstly, the participants were allocated to the respective study group (intervention or control) on a free-decision basis. This procedure can be justified by the local restrictions in association with the COVID-19 pandemic and the consecutive necessity to initiate the study with the first group of participants who were then located to the control group due to an immediate local lockdown order. This selection bias which can be also justified by relatively small sample size in this pilot study also caused an imbalance of group sizes and baseline characteristics of our study. Secondly, we did not perform a cardio-pulmonary exercise test during the screening visit but only after the two diet periods by the fact that CPX- testing is a time-consuming procedure and study-related interventions had to be kept at a minimum in this unfunded and timely limited study. For this reason, we were not able to compare interindividual physical performance data during baseline and after the respective nutritional interventions. Furthermore, we did not perform gender-specific analyses as most participants within the intervention group and only one person of the control group was female. Finally, it must be considered that any kind of diet might be impacting psychological factors such as quality of life [36]; unfortunately, this outcome was not assessed in our study.

## 5. Conclusions

An episode of three weeks of HC and subsequent LC diet altered various markers of physical performance in favor of the HC diet as given via longer TTE and higher P_peak_. Markers of body composition were altered by both diet interventions to a similar extent. Significant reductions in total and LDL-cholesterol were demonstrated in response to the HC diet, while triglycerides increased after the HC diet.

## Figures and Tables

**Figure 1 nutrients-14-00423-f001:**
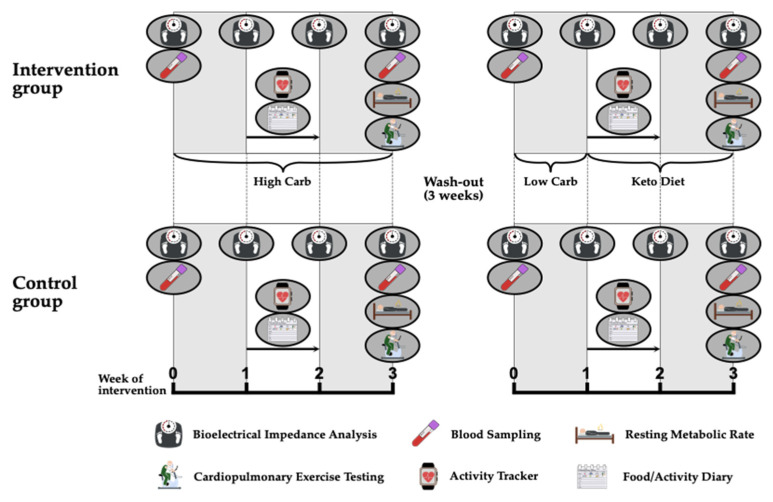
Study procedures.

**Figure 2 nutrients-14-00423-f002:**
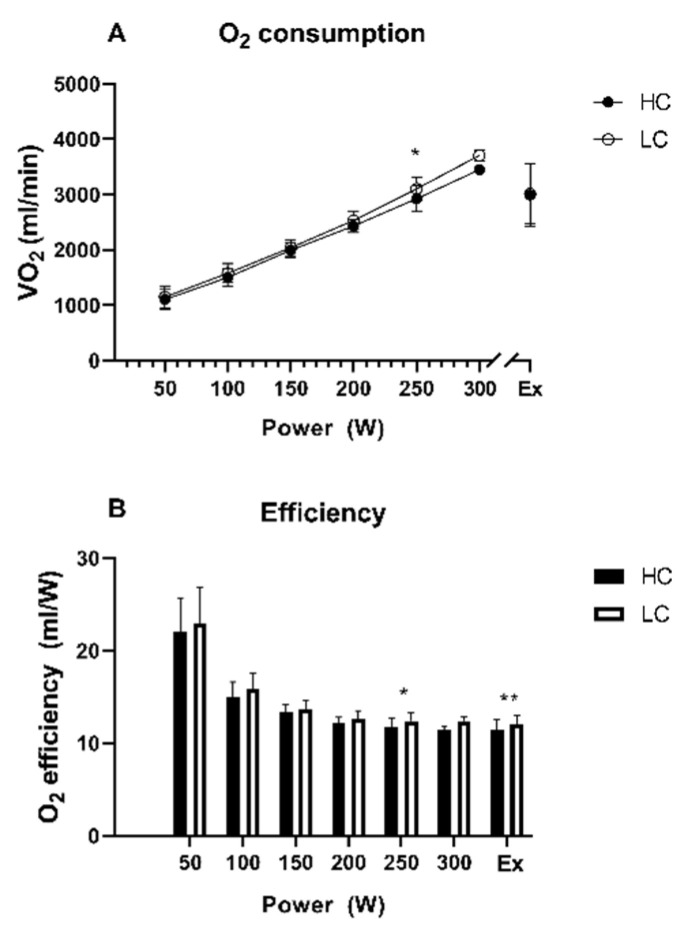
Course over time for O_2_ consumption (**A**) and O_2_ efficiency (**B**) during cardio-pulmonary exercise testing. Ex = time point of exhaustion, HC = high-carb, LC = low-carb, * indicates *p* < 0.05 between HC and LC group, ** indicates *p* < 0.01 between HC and LC group.

**Figure 3 nutrients-14-00423-f003:**
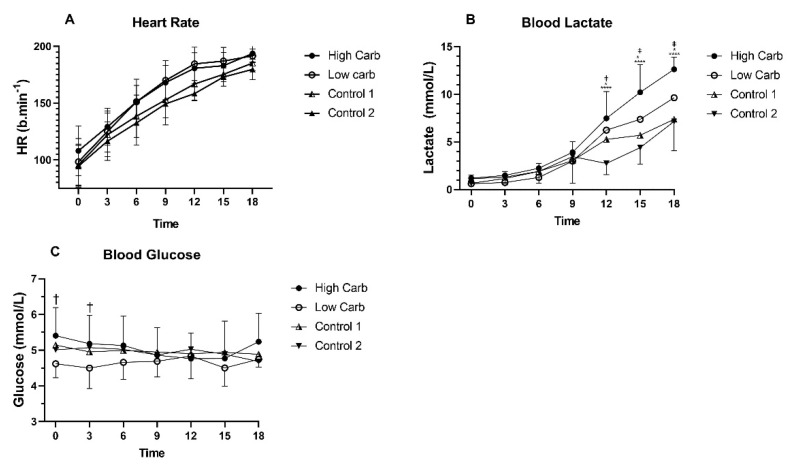
Course over time for heart rate (**A**), lactate (**B**) and glucose (**C**) during cardio-pulmonary exercise testing. * Indicates *p* < 0.05 between HC and control group, **** indicates *p* < 0.0001 between HC and control group. Control 1 and 2 indicate the respective 3-week episodes of the control group. † Indicates *p* < 0.05 between HC and LC. ‡ Indicates *p* = 0.05 between LC and control group.

**Figure 4 nutrients-14-00423-f004:**
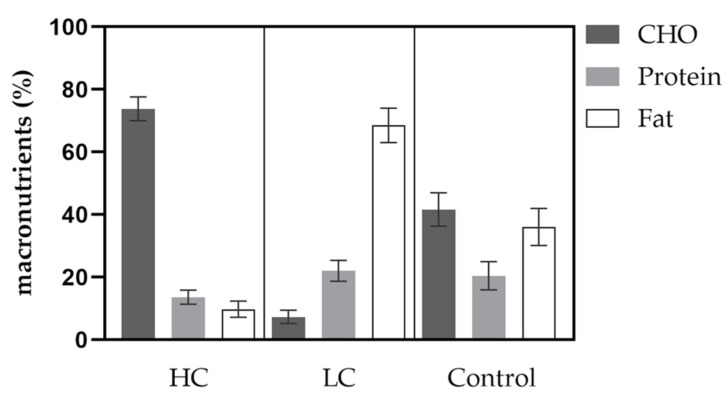
Distribution of macronutrients according to diet groups and control. CHO = carbohydrate.

**Table 1 nutrients-14-00423-t001:** Baseline characteristics of the investigated study cohort.

	Intervention Group (*n* = 18)	Control Group (*n* = 6)	*p*-Value
Females	13	1	
Age (years)	24.9 ± 1.3	28.5 ± 6.9	0.03
BMI (kg/m^2^)	21.8 ± 1.8	23.1 ± 2.9	0.22
CRP (mg/dL)	1.2 ± 0.7	1.4 ± 1.0	0.09
HDL-C (mg/dL)	76.7 ± 9.7	58.7 ± 11.2	<0.0001
LDL-C (mg/dL)	95.8 ± 30.7	87.1 ± 13.7	0.54
Cholesterol (mg/dL)	188 ± 34	163 ± 26	0.09
Triglycerides (mg/dL)	78 ± 32	84 ± 37	0.30
BHB (mmol/L)	0.04 ± 0.04	0.09 ± 0.01	0.44

BM = body mass; BMI= body mass index; CRP = C-reactive protein; HDL-C = high-density lipoprotein; LDL-C = low-density lipoprotein, BHB = Betahydroxybutyrate.

**Table 2 nutrients-14-00423-t002:** Performance parameters during different time points of investigation.

Parameter	After HC	After LC	After Control 1	After Control 2	*p*-Value HC vs. LC	*p*-Value HC vs. Control 1	*p*-Value HC vs. Control 2	*p*-Value LC vs. Control 1	*p*-Value LC vs. Control 2
VO_2peak_ (mL/min/kg)	47 ± 7	47 ± 7	49 ± 7	47 ± 9	ns	ns		ns	ns
P_Peak_ (Watt)	251 ± 43	240 ± 45	311 ± 61	308 ± 69	0.0001	0.02	0.03	0.009	0.02
TTE (Minutes)	14.47 ± 2.36	14.08 ± 2.36	18.20 ± 3.42	18.1 ± 4.1	0.002	0.02	0.04	0.01	0.009

The *p*-values indicate significance according to the change when compared to the other groups. Control 1 and 2 indicate the respective 3-week episodes of the control group.

**Table 3 nutrients-14-00423-t003:** Body composition during high-carbohydrate (HC) and low-carbohydrate (LC) diet.

	HC					LC				
Parameter	Baseline	Week 1	Week 2	Week 3	Δ Week 3	Baseline	Week 1	Week 2	Week 3	Δ Week 3
Body mass (kg)	65.2 ± 11.1	64.6 ± 11.0 **	63.8 ± 10.9 ***	63.8 ±11.1 ***	−1.4 ± 0.9	64.8 ± 11.6	63.9 ± 11.2 ***	63.5 ± 11.1 ***	63.5 ±11.3 ***	−1.3 ± 0.9
Lean body mass (kg)	50.5 ± 11.7	50.5 ± 11.7	50.2 ± 11.7	50.4 ±11.5	−0.2 ± 0.9	50.5 ± 11.7	50.4 ± 11.7	50.3 ± 11.5	50.6 ± 11.7	0.2 ± 1.1
Skeletal muscle mass (kg)	28.3 ± 7.3	28.3 ± 7.3	28.1 ± 7.2	28.2 ± 7.0	−0.1 ± 0.6	28.3 ± 7.2	28.3 ± 7.2	28.2 ± 7.1	28.4 ± 7.2	0.1 ± 0.7
Body fat (%)	22.7 ± 6.5	22.1 ± 6.5	21.6 ± 6.6 **	21.2 ± 6.2 **	−1.4 ± 1.4	22.3 ± 5.7	21.3 ± 6.0 ***	21.0 ± 6.1 ***	20.6 ± 6.0 ***	−1.7 ± 1.4
Visceral fat (cm^2^)	58.6 ± 17.8	56.3 ± 16.8 *	54.1 ± 16.8 ***	53.9 ± 16.2 ***	−4.7 ± 3.8	57.6 ± 15.9	53.3 ± 16.3 ***	51.9 ± 16.5 ***	50.6 ± 16.6 ***	−7.0 ± 4.6
Total body water (L)	37.2 ± 8.6	37.1 ± 8.6	37.0 ± 8.6	37.1 ± 8.5	−0.1 ± 0.7	37.1 ± 8.6	37.1 ± 8.6	37.0 ± 8.5	37.2 ± 8.7	0.1 ± 0.8

Significant differences to baseline values are indicated with * *p* <0.05; ** *p* <0.01; *** *p* <0.001.

**Table 4 nutrients-14-00423-t004:** Laboratory parameters during different time points of investigation.

Parameter	HC Baseline	after HC	*p*-Value Baseline vs. after HC	LC Baseline	after LC	*p*-Value Baseline vs. after LC	Control 1 Baseline	after Control 1	*p*-Value Baseline vs. Control 1	Control 2 Baseline	after Control 2	*p*-Value Baseline vs. Control 2
BHB (mmol/L)	0.06 ± 0.05	0.05 ± 0.07	ns	0.04 ± 0.02	0.42 ± 0.27	<0.0001	0.09 ± 0.1	0.05 ± 0.02	ns	0.07 ± 0.07	0.05 ± 0.02	ns
Total cholesterol (mg/dL)	189 ± 34	158 ± 27	0.02	187 ± 35	203 ± 60	ns	163 ± 26	154 ± 35	ns	159 ± 41	187 ± 43	ns
LDL-C (mg/dL)	98 ± 34	78 ± 23	0.02	94 ± 28	105 ± 51	ns	87 ± 27	77 ± 30	ns	85 ± 37	110 ± 41	ns
HDL-C (mg/dL)	77 ± 9	58 ± 9	<0.0001	77 ± 11	82 ± 16	ns	59 ± 11	56 ± 11	ns	58 ± 17	59 ± 11	ns
TG (mg/dL)	76 ± 38	104 ± 44	0.005	81 ± 26	80 ± 24	ns	86 ± 45	104 ± 45	ns	83 ± 31	96 ± 33	ns
CRP mg/dL)	1.3 ± 1.4	0.9 ± 0.4	ns	0.9 ± 0.8	0.8 ± 0.4	ns	1.4 ± 1.0	1.1 ± 0.7	ns	2.0 ± 1.7	1.9 ± 2.3	ns
IL-6 (mg/dL)	1.6 ± 0.2	1.7 ± 0.9	ns	1.7 ± 0.4	1.6 ± 0.2	ns	1.8 ± 0.4	1.9 ± 0.4	ns	1.9 ± 0.7	1.5 ± 0.1	ns

The *p*-values indicate significance according to the change when compared to baseline. Control 1 and 2 indicate the respective 3-week episodes of the control group. BHB = Betahydroxybutyrate; LDL = low-density-lipoprotein; HDL = high-density-lipoprotein; CRP = C-reactive protein; IL-6 = Interleukin-6; TG = triglycerides.

## Data Availability

Not applicable.

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
