# Peer review of "The Impact of a High-Carbohydrate/Low Fat vs. Low-Carbohydrate Diet on Performance and Body Composition in Physically Active Adults: A Cross-Over Controlled Trial"

_nutrients, 2022, doi:10.3390/nu14030423_

Round 1

Reviewer 1 Report

The study findings are interesting and meaningful to suggest the effects of high-carbohydrate and low-carbohydrate diets on performance and body composition in physically active adults. However, as the authors indicated, non-randomized design and small sample size seem important limitations of this study despite the authors’ efforts. Increasing sample size will greatly help the quality of the study. In addition, the authors may need to consider the following questions and comments to improve the manuscript.

Baseline characteristics of intervention and control groups should be similar, but sex and age distribution

Did the authors provide any direction, counseling, or education to make study participants follow a high-carbohydrate or low-carbohydrate diet?

How was the compliance of study participants?

It would be more understandable, if some of results such as body composition and biochemical parameters are presented in tables including differences between high-carbohydrate and low-carbohydrate diets.

Author Response

Thank you very much for reviewing our manuscript entitled “The impact of a high-carbohydrate vs. low-carbohydrate diet on performance and body composition in physically active adults: a controlled cross-over trial”. Please find below a point-to-point response to the specific comments.

The study findings are interesting and meaningful to suggest the effects of high-carbohydrate and low-carbohydrate diets on performance and body composition in physically active adults. However, as the authors indicated, non-randomized design and small sample size seem important limitations of this study despite the authors’ efforts. Increasing sample size will greatly help the quality of the study. In addition, the authors may need to consider the following questions and comments to improve the manuscript.

First, we would like to thank the reviewer for assessing our manuscript and addressing important points to increase the quality of our work. As stated correctly by the reviewer, we would have had much appreciated performing this trial as an RCT, however, due to the pandemic and its accompanied restrictions, we were not able to conduct our trial as planned. Furthermore, the recruitment procedure and hence the small number of participants were influenced by the pandemic, since potential participants were afraid to visit our research facility. Taking these two aspects together, we hope the reviewer can understand that we tried our best to perform this study.

Baseline characteristics of intervention and control groups should be similar, but sex and age distribution

Thank you very much for the comment on which we absolutely agree. As stated in the manuscript, this study was designed as pilot trial characterized by a relatively small sample-size. We mainly focused on the inter-individual changes in response to the different diets in this cross-over study. Comparisons with the control group were done with caution, since we were aware of the non-equally distributed participant characteristics at baseline. In addition, as also mentioned in the manuscript, the allocation of the participants to the groups (control or intervention) was highly impacted by sudden COVID-related lockdown restrictions, which made a formal randomization procedure almost impossible. The issue with the heterogeneously distributed participant characteristics has already been specified as a limitation in the first version of the manuscript. In addition, in the updated manuscript, we mentioned the small sample-size as limitation.

This selection bias which can be also justified by a relatively small sample size in this pilot study also caused an imbalance of group sizes and baseline characteristics of our study.”

Did the authors provide any direction, counseling, or education to make study participants follow a high-carbohydrate or low-carbohydrate diet?

We want to thank the reviewer for this important comment. We added now details about the dietary procedures and nutritional recommendations to adhere to the specific diet.

“The high carbohydrate content was achieved with complex carbohydrates, such as found in whole meal products, potatoes or brown rice, while carbohydrates consumed via sucrose and fructose were mainly avoided. After a wash-out period of approximately 3 weeks, the second intervention period started with a one-week lead-in-phase of a LC diet (20-25% carbohydrates, 15% proteins, 60 - 65% fat) followed by a 2-week ketogenic diet (5-7% carbohydrates, 15% proteins, 80% fat). The diet consisted mainly of fish, meat, nuts, vegetables and dairy products. Participants were educated by a nutritionist what to eat during the HC and LC diets and example menus were provided.”

How was the compliance of study participants?

Thank you very much for this question. The participants´ compliance to adhere to the study diets can be considered as highly satisfying. Figure 4 which describes the consumed macronutrient intake of the total study population indicates an appropriate achievement of the respective macronutrient proportion which was assessed by means of food diaries. This high compliance rate was also physiologically confirmed, since betahydroxbutyrate levels were significantly elevated after the LC diet (and not after HC diet).

 It would be more understandable, if some of results such as body composition and biochemical parameters are presented in tables including differences between high-carbohydrate and low-carbohydrate diets.

Thank you for this useful comment. We additionally inserted a table including the body composition parameters

Reviewer 2 Report

General Comments

The objective of the study submitted by Wachsmuth et al. was to assess the effects of a high versus a low-carbohydrate diet in healthy individuals on physical performance, body composition and blood metabolites. The paper is well written and addresses an important question in the field. It would have been more informative if the implication of the study was further clarified in the introduction. The M&M section needs further details as some of them are highlighted in my specific comments. In results section, some information is missing and need to be added in the paper (please see the specific comments). Some of important results are not fully discussed and need further discussions as I have mentioned below.

Specific Comments

Lines 20-22: this part is still “Methods”

Line 79-80: please provide a reference

Lines 118-122: please provide the diets ingredients

Line 123: how much was the calorie intake?

Lines 135-137: please provide a reference or justification for this method

Line 148-149: what was those standardized room temperature and humidity?

Line 171: serum or plasma?

Lines 172-174: no information has been given how these assays were done. A separate section for these analyses should be added in the M&M section before statistical analysis.

Line 296: please add the error bars for Fig. 4.

Line 38: please add the unit for BHB. I think the results of this part (Table 2) will be more clear if presented as graphs rather than a chart.

Line 373-386: Why total cholesterol and LDL-cholesterol decreased after the HC diet? Why no changes were seen with LC diet? Please add more discussions.

Lines 421-436: Authors have used HC and low fat vs. LC and high fat diets. Will the results be different if protein instead of fat were manipulated? What is the contribution of dietary fat in obtained results? I would suggest clarifying that in the paper discussion and if needed use the term “high carbohydrate-low fat diet” and “low carbohydrate-high fat diet”.

Author Response

General Comments

The objective of the study submitted by Wachsmuth et al. was to assess the effects of a high versus a low-carbohydrate diet in healthy individuals on physical performance, body composition and blood metabolites. The paper is well written and addresses an important question in the field. It would have been more informative if the implication of the study was further clarified in the introduction. The M&M section needs further details as some of them are highlighted in my specific comments. In results section, some information is missing and need to be added in the paper (please see the specific comments). Some of important results are not fully discussed and need further discussions as I have mentioned below.

We highly appreciate the thorough review and the positive feedback provided by this reviewer.

Specific Comments

Lines 20-22: this part is still “Methods”

Thank you very much for the comment. We agree with the reviewer that information on the participants belongs to the methods paragraph. This is now amended accordingly.

Line 79-80: please provide a reference

Thank you for requesting a reference to clarify this. This is now provided in the updated manuscript. doi.org/10.1093/ajcn.81.2.341

Lines 118-122: please provide the diets ingredients

Based on the reviewers` comment, we have added a clearer description of recommended ingredients to achieve the specific macronutrient amount.

“The high carbohydrate content was achieved with complex carbohydrates, such as found in whole meal products, potatoes or brown rice, while carbohydrates consumed via sucrose and fructose were mainly avoided. After a wash-out period of approximately 3 weeks, the second intervention period started with a one-week lead-in-phase of a LC diet (20-25% carbohydrates, 15% proteins, 60 - 65% fat) followed by a 2-week ketogenic diet (5-7% carbohydrates, 15% proteins, 80% fat). The diet consisted mainly of fish, meat, nuts, vegetables and dairy products. Participants were educated by a nutritionist what to eat during the HC and LC diets and example menus were provided.”

Line 123: how much was the calorie intake?

Thank you very much for the comment. Based on the reviewers’ request, we refer to section 3.2.6., which describes the nutrition during the specific diets as assessed by food diaries.

Lines 135-137: please provide a reference or justification for this method

We have added a reference including the used CPX protocol in the manuscript: DOI: 10.4061/2011/209302.

Line 148-149: what was those standardized room temperature and humidity?

We want to thank the reviewer for the comment. In the updated version of the manuscript (method section), we added information about standardized humidity and temperature of our laboratory as following: “The temperature and environmental humidity in the laboratory of the research facility were stable during all visit days with 24°C and 50%, respectively.”

Line 171: serum or plasma?

We thank the reviewer for making us aware about this lack of clarity – we analyzed serum samples.

Lines 172-174: no information has been given how these assays were done. A separate section for these analyses should be added in the M&M section before statistical analysis.

Thank you for your suggestion to clarify the assay used for blood analysis. This is now added as follwing “All the listed measurements were performed on a cobas 8000 analyzer (Roche Diagnostics) with standardized assays by the same manufacturer which are calibrated to international standards.”

Line 296: please add the error bars for Fig. 4.

We want to thank the reviewer for pointing out this inaccuracy. In the updated version of our paper, we added error bars in figure 4.

Line 38: please add the unit for BHB. I think the results of this part (Table 2) will be more clear if presented as graphs rather than a chart.

We want to thank the reviewer for this comment. We added the unit for BHB in the Table. We agree with the concern about low-quality tables. Based on the reviewers’ suggestion, we inserted frame lines in the tables, which makes it much clearer to read. 

Line 373-386: Why total cholesterol and LDL-cholesterol decreased after the HC diet? Why no changes were seen with LC diet? Please add more discussions.

Thank you for bringing this point. We have added some more declarations for these changes in lipids in the discussion part.

Apart from a significant but though physiological rise in betahydroxybutyrate in the LC diet group, which implicates sufficient adherence to the diet measures, some alterations in the lipid profiles were observed. In our participants, who were all classified as normocholesterolemic, total cholesterol and LDL-C decreased significantly during the HC intervention, which is in line with previous research that found reductions in total and LDL cholesterol when normocholesterolemic participants switched to a diet low in saturated fat but high in carbohydrates [27]. In contrast, total cholesterol and LDL-C numerically increased during the LC diet without reaching statistical significance. This finding confirms results from a recent study, in which a LC/high fat diet increased LDL- cholesterol by 44% with high interindividual variability of increase (5-107%) during a 4-week diet intervention [28]. The only slight increase of cholesterol in our study might be due to the fact that the study was conducted in well-trained individuals who regularly perform physical activity/exercise. The effect of this physically active lifestyle might attenuate the cholesterol increase induced by the LC/high carb diet [29]. However, as seen for example in the DIRECT study, the effect of LC diet on LDL-C is changing over time, hence the duration of the study seems to be an important factor to consider when evaluating the lipid effects of LC diets [30]. In our study, triglyceride levels significantly increased after the HC diet and tended to decrease without significance during the LC diet. The effect of different diets on triglyceride levels remain heterogenous, although a reduction in triglycerides with LC is suggested in a meta-analysis in people with diabetes mellitus type 2 [3] and the phenomenon of carbohydrate-induced hypertriglyceridemia (HPTG) has been previously observed in studies where HC and low-fat diets were investigated.

Lines 421-436: Authors have used HC and low fat vs. LC and high fat diets. Will the results be different if protein instead of fat were manipulated? What is the contribution of dietary fat in obtained results? I would suggest clarifying that in the paper discussion and if needed use the term “high carbohydrate-low fat diet” and “low carbohydrate-high fat diet”.

Thank you for this appreciated comment. Not only the carbohydrate but also the proportion of the other macronutrients (proteins [15% in both arms]) and fat [5-10% in the HC group, 80% in the LC group] was predetermined and standardized due to our protocol (see section 2.2. diet intervention). In order to clarify this, we have specified the abbreviation HC meaning “high carb - low fat” also in the title.

Due to the well adherence to the respective diets seen in our study, this study cannot explain whether a more variable proportion of consumed protein or fat might influence the results. 

Round 2

Reviewer 1 Report

Thank you for addressing my previous comments/questions. I understand that some limitations were unavoidable due to the pandemic.

I just have a minor comment. Both high-carbohydrate and low-carbohydrate diets diets showed similar effects on anthropometric measurements. A little discussion about potential reasons for the similar effects of the two diets would be helpful.

Author Response

Dear Reviewer,

Thank you very much for the re-evaluation of our manuscript – we do much appreciate. Please find below the response to the specific query.

I just have a minor comment. Both high-carbohydrate and low-carbohydrate diets showed similar effects on anthropometric measurements. A little discussion about potential reasons for the similar effects of the two diets would be helpful.

Thank you for this specific comment. Indeed, in our study, both diets followed for 3 weeks, respectively, contributed to significant weight loss in a similar extent. Following the advice of the reviewer, we added the following paragraph within the discussion section to shed some light on this finding:

Existing evidence on changes in anthropometric data in response to HC and LC diets remain highly heterogenous and an active matter of debate. The partly confusing and discrepant results in regard to diet-induced weight effects gained from clinical trials might be justified by different durations of dieting, variable definitions of high and low carb diets as well as different populations which were investigated. In our point of view, each broadly accepted diet dedicated to reduce weight has the potential to positively modify weight control when conducted as recommended in particular when food diaries are kept and nutritional support by experts is guaranteed as it was the case in our study. Considering this, a crucial point in dieting successfully includes a positive encouragement and an individual interest in strictly adhering to the suggested nutritional behavior. Intriguingly, when evaluating the body weight reduction, the diet-unspecific “adherence- and support-effects” might overcome the disbalanced kcal/day intake in comparison of both diet interventions; even though within the HC arm less kcal/day were consumed, the bodyweight reduction was nearly congruent for HC and LC. From our point of view, a generalized diet prescription to reduce bodyweight by means of HC or LC might be outdated and the focus should be shifted preferably towards a personalized dietary prescription. 

Reviewer 2 Report

The authors have addressed the majority of my concerns and have applied the necessary changes in the revised manuscript.

Author Response

We again express our thanks to this reviewer.